# Predictive and Prognostic Value of BRAF and NRAS Mutation of 159 Sentinel Lymph Node Cases in Melanoma—A Retrospective Single-Institute Study

**DOI:** 10.3390/cancers13133302

**Published:** 2021-06-30

**Authors:** Gabriella Liszkay, Zoltán Mátrai, Kata Czirbesz, Nóra Jani, Eszter Bencze, István Kenessey

**Affiliations:** 1Department of Dermato-Oncology, National Institute of Oncology, 1122 Budapest, Hungary; czirbikatu@gmail.com; 2Department of Breast and Sarcoma Surgery, National Institute of Oncology, 1122 Budapest, Hungary; matraidok@oncol.hu; 3Department of Surgical and Molecular Pathology, National Institute of Oncology, 1122 Budapest, Hungary; nora.jani.dr@gmail.com (N.J.); ri3lity@gmail.com (E.B.); 4National Cancer Registry, National Institute of Oncology, 1122 Budapest, Hungary; kenessey.istvan@oncol.hu; 52nd Department of Pathology, Semmelweis University, 1085 Budapest, Hungary

**Keywords:** melanoma, sentinel lymph node, NRAS, BRAF, progression

## Abstract

**Simple Summary:**

Sentinel lymph node (SLN) status is still the most important prognostic factor for melanoma patients; however, the efficacy of completing lymph node dissection remains questionable. The aim of our study was to assess the correlation between known prognostic factors, mutational occurrence of BRAF and NRAS in the primary tumor, and SLN status. Statistical analysis revealed that Breslow thickness was associated with SLN status; however, neither NRAS nor BRAF showed a predictive value. Furthermore, NRAS mutation in primary tumors proved to be an independent factor of tumor progression. This suggests that regardless of the SLN status, the NRAS-mutant subgroup of patients requires closer monitoring.

**Abstract:**

Purpose: To assess the prognostic role of sentinel lymph node status (SLN) in melanoma patients, a statistical comparison was performed with the application of already known prognostic factors, mutational occurrence of BRAF and NRAS in the primary tumor, as well as disease outcome. Methods: Our retrospective single-center study involved 159 melanoma cases, who underwent SLN biopsy. The following clinico-pathological data were collected: age, gender, location of primary tumor, Breslow thickness, ulceration degree, histological subtype, mitosis count, lymphovascular and perineural invasion, presence of tumor-infiltrating lymphocytes, regression signs, mutations of BRAF and NRAS of the primary tumors, and SLN status. Results: From the studied clinico-pathological factors, only Breslow thickness increased the risk of SLN positivity (*p* = 0.025) by multivariate analysis, while neither BRAF nor NRAS mutation of the primary tumor proved to be a predictor of the SLN status. While the NRAS-mutant subgroup showed the most unfavorable outcome for progression-free and distant metastasis-free survival, their rate of positive SLNs proved to be relatively lower than that of patient groups with BRAF mutation and double-wild-type phenotypes. Conclusion: Similarly to the importance of SLN positivity, NRAS mutation of the primary tumor proved to be an independent prognostic factor of progression. Therefore, despite negative SLN, this NRAS-mutant subgroup of patients still requires closer monitoring to detect disease progression.

## 1. Introduction

The incidence of melanoma shows a continuous increase worldwide. During the past few decades, the number of new cases has multiplied [1,2]. Experimental and clinical research in the last decades investigated the molecular and immunological background of melanoma and successfully established the base of novel innovative therapeutic modalities such as target-based therapies and immunotherapies and the clinical application of new prognostic and predictive factors [3,4,5].

Oncogenic mutations in *BRAF* and *NRAS* proved to be the most common genetic alterations in cutaneous melanoma (40–60% and 15–20%, respectively), while the *KIT* gene is frequently affected in mucosal melanomas (5–40%), and the majority of uveal melanomas harbors mutations of *GNAQ* or GNA-11 (80%) [6,7].

Somatic mutations of *BRAF* and *NRAS* result in the hyperactivation of mitogen-activated protein kinase (MAPK) signaling that drives tumor growth and leads to progression of the disease [8,9].

In 90% of cases with a BRAF mutation, a valine-to-glutamic-acid mutation is present at codon 600 of exon 15 (V600E). The development of specific inhibitors, such as vemurafenib, opened a new horizon for melanoma therapy. However, other known rare BRAF mutations also appear (e.g., V600K, V600R, V600D). BRAF mutations more frequently occur at younger ages and on trunk location and are associated with chronic UV exposure [10]. Although the presence of BRAF mutation is an attractive target of melanoma therapy, its prognostic value is still elusive.

*RAS* was the first discovered oncogene in melanoma. The incidence of *RAS* mutation is approximately 20%, with the majority found in *NRAS*, while mutations of *KRAS* and *HRAS* may occur in 1–2% of cases. Mostly, a glutamine–leucin substitution is detected in exon 3, codon 61 position, while alterations of exon 2, codons 12 and 13 are relatively rare [11,12]. Similarly to BRAF, it seems that high UV exposure induces the development of NRAS mutations as well. Unfortunately, specific molecular therapy against NRAS-mutated melanoma has not been accepted yet. Previously, based on clinical trials, some therapeutic advantages of a MEK inhibitory strategy were reported [13]. Furthermore, studies suggested that immunotherapy was more effective in patients with NRAS mutation, but some studies conclude that NRAS mutation in melanoma has a negative impact on disease outcome [14].

Although in sentinel lymph node (SLN)-positive cases the efficacy of completing lymph node dissection remains questionable [15], the role of sentinel lymph node biopsy is inevitable for regional staging and therapy designing, and sentinel lymph node status is one of the most important prognostic factors in melanoma.

The aim of our study was to assess the correlation between known prognostic factors of melanoma, mutational occurrence of BRAF and NRAS in the primary tumor, and sentinel lymph node status. Moreover, we investigated the association of these factors with disease outcome.

## 2. Materials and Methods

### 2.1. Patients

Our retrospective single-center study involved 159 patients who were surgically treated for melanoma at the National Institute of Oncology (Budapest, Hungary) between October 2011 and July 2015. From the institutional database, the following clinico-pathological data were collected: age, gender, location of the primary tumor, Breslow thickness, ulceration, histological subtype, mitosis count, lymphovascular and perineural invasion, evaluation of tumor-infiltrating lymphocytes, signs of regression, and SLN status. Microscopic satellitosis was not included in the analysis, because of the low number (less than 10) of cases. Primary melanomas were removed in two steps; a 5 mm safety margin was completed to 1–2 cm according to the WHO guideline (no insufficient margins were found in the population of patients.) SLN biopsy was performed in the following cases: intermediate tumor thickness (1–4 mm), less than 1 mm if the primary tumor was ulcerated, Clark level higher than III, lymphovascular invasion, or high mitosis activity, more than 4 mm Breslow thickness if the tumors were not ulcerated. SLN biopsy was performed by the double-labeling technique 4–8 weeks after primary surgery. In cases of positive SLNs, complete regional lymph node dissection (RLND) was indicated. SLNs were histologically investigated in serial sections stained with hematoxylin and eosin. In addition, HMB45, S100, and Melan-A immunohistochemistry was performed to confirm the SLN status, as we previously reported [16].

Patients with negative SLNs received low-dose interferon-α therapy for 18 months. In positive-SLN cases, intermediate- or high-dose interferon-α therapy was indicated after regional block dissection. According to the decision of a multidisciplinary oncoteam, for cases with a disseminated tumor, targeted, immune, and chemotherapy were performed. Regression of primary tumor was classified as lower or higher than 75% and late or early, similarly to the protocol of the College of American Pathologists [17].

Follow-up data were obtained from the institutional database and National Cancer Registry of Hungary. The follow-up period ended in October 2019.

### 2.2. Genetic Subtyping

For further analysis, molecular categorization was performed. Genomic DNA was isolated from formalin-fixed, paraffin-embedded tissue (FFPET) using the cobas^®^ DNA Sample Preparation kit (Roche Diagnostics, Basel, Switzerland). The target DNAs were amplified and detected on the cobas z 480 analyzer using the amplification and detection reagents provided in the BRAF/NRAS Mutation Test (LSR) kit (Roche Diagnostics, Basel, Switzerland).

BRAF/NRAS Mutation Test (LSR) uses primers that define specific base-pair sequences for each of the targeted mutations. Amplification occurs only in the regions of the *BRAF* or *NRAS* genes between the primers; the entire gene is not amplified. *BRAF* sequences range from 101 to 120 base pairs. *NRAS* sequences range from 94 to 121 base pairs.

The test is designed to detect the following mutations (*n* = 36) at a percent mutation of 5% or greater:V600E, V600E2, V600D, V600K, V600R, and K601E in *BRAF* exon 15G466A, G466V, G469A, G469R, and G469V in *BRAF* exon 11G12A, G12C, G12D, G12R, G12S, G12V, G13A, G13C, G13D, G13R, G13S, G13V, and A18T in *NRAS* exon 2A59D, A59T, Q61Ht, Q61Hc, Q61K, Q61L, Q61P, and Q61R in *NRAS* exon 3K117Nc, K117Nt, A146T, and A146V in *NRAS* exon 4

### 2.3. Statistical Analysis

Numeric parameters were compared by the Mann–Whitney or Kruskal–Wallis test with post hoc analysis. Categorical data were analyzed by the Chi-square test or Fisher’s exact probability test. Survival periods were determined as the time period from the date of SLN biopsy to the date of the last visit or defined complete event (death, progression, distant metastasis). Thus, overall survival (OS), disease-specific survival (DSS), progression-free survival (PFS), and distant metastasis-free survival (DMFS) were calculated. Survival analyses were done using the Kaplan–Meier method and log-rank statistics. Univariate and multivariate analyses of prognostic factors were done using the Cox’s regression model. Probability of sentinel lymph node positivity was assessed by binary logistic regression model. Differences were considered statistically significant when the *p*-value proved to be lower than 0.05. All statistical calculations were performed by Statistica 13.4 (TIBCO Software, Palo Alto, CA, USA).

### 2.4. Ethical Permission

The study was conducted under the ethical permission of the Scientific and Research Ethics Committee of the Medical Research Council (approval number: 15140/2017) and was carried out in accordance with The Code of Ethics of the World Medical Association (Declaration of Helsinki) for experiments involving humans.

## 3. Results

The median follow-up period of the studied 159 patients was 61 months (range: 1–96 months). The median age was 59 years (range: 18–83 years). Out of 159 patients, 71 were male (44.7%), and 88 were female (55.3%). The most frequent location of the primary tumor was the trunk (77; 48.4%), followed by the lower extremities (47; 29.6%), then the upper extremities (35; 22%). Median Breslow thickness was 1.8 mm (range: 0.51–20 mm). Forty-nine of the primary tumors were ulcerated (30.8%). The most frequent histological subtype was superficial spreading melanoma (SSM, 124; 78%); 31 cases (19.5%) presented with nodular melanoma (NM), while in 4 cases (2.5%), other histological types were identified (Table 1).

The median mitosis count was 4/mm^2^ (range: 0–31); in 132 tumors (83%) neither lymphovascular nor perineural invasion was detected. In 86 cases (54.1%), tumor-infiltrating lymphocytes (TIL) were detected. Early regression of the primary tumor occurred in 9 patients (5.7%), late regression of less than 75% occurred in 50 tumors (31.4%), while 75% regression was exceeded in 21 patients (13.2%). The primary tumor of 90 patients harbored a BRAF mutation (56.6%), which was V600E in 87%, V600K in 10%, and V600R in 3% of the patients. An NRAS mutation was detected in 29 patients (18.2%), specifically, in exon 3 codon 61 (97%) and exon 2 codon 13 (3%); 28 patients proved to belong to the double-wild-type group (17.6%), and genotyping was not available for 12 patients (7.5%). The SLN status was negative in 130 cases (81.8%), while 29 cases showed SLN positivity (18.2%).

The SLN status showed a positive association with Breslow thickness of the primary tumors (*p* = 0.008). In SLN-negative cases, the median Breslow thickness was 1.64 mm (range: 0.51–20 mm), whereas in positive cases, it was 2.45 mm (range: 0.79–15 mm). SLN status and ulceration of the primary tumor showed significant association as well (*p* = 0.007): 26.2% of the studied tumors were ulcerated in negative-SLN cases, and 51.7% in positive-SLN cases. Additionally, mitosis count of the primary tumor differed between SLN-negative and -positive groups (*p* = 0.009), corresponding to 3/mm^2^ (range: 0–31) and 5/mm^2^ (1–30), respectively.

Obviously, the presence of lymphovascular and/or perineural invasion of the primary tumor was associated with SLN status (*p* = 0.004); it was present in 34.5% of SLN-positive cases and in only 12.3% of the negative cases. A slightly higher occurrence of BRAF mutation was detected in the primary tumors of SLN-positive patients with respect to the SLN-negatives cases (65.5% and 54.6%, respectively); however, the difference was not statistically significant. Interestingly, NRAS mutation of the primary tumor was present only in 6.9% of SLN-positive tumors, while it was detected in 20.8% of SLN-negative cases; however, this difference was not statistically significant either (Table 2).

When analyzing the correlation of different prognostic factors with SLN status, a multivariate analysis revealed that the only significant parameter was Breslow thickness of the primary tumor (OR: 4.222; 95%; CI: 1.201–14.873; *p* = 0.025), while the other studied variables did not affect the risk of sentinel node positivity (Table 3).

When evaluating the parameters of BRAF, NRAS mutant, and double-wild-type primary tumors, significant differences were revealed for age and Breslow thickness of primary tumor (*p* = 0.001 and *p* = 0.018, respectively). BRAF-mutant patients were younger, NRAS-mutant primary tumors were thicker than those in the other two groups. Trunk location was slightly more frequent, found in 58.6% of NRAS-mutant, 51.1% of BRAF-mutant, and 31.1% of double-wild-type tumors. We also observed that 24.1% of NRAS-mutant tumors proved to be nodular melanoma, while 16.7% of BRAF-mutant and 17.9% of double-wild-type tumors belonged to that histological category (Table 4).

At the last follow-up, 130 patients were alive (81.8%), and 29 had died (18.2%). A total of 123 patients (77.4%) were tumor-free, 7 received innovative therapies due to metastatic disease (4.4%). In 29 cases (18.2%), progression of the disease was detected, i.e., locoregional progression in 7 patients (4.6%) and distant metastasis in 22 cases (13.8%), respectively. Out of the 22 distant-metastatic cases, 9 had previous locoregional progression.

The univariate Cox proportional hazard model confirmed previously reported findings that Breslow-thickness of the primary tumor, ulceration degree, mitosis level, and an invasive spreading pattern highly affected every survival endpoint (Table 5), while patients’ age affected DMFS and OS. In addition, sentinel lymph node status was associated with the risk of progression (*p* = 0.001) and distant metastasis-free survival (*p* = 0.006), but not with OS and DSS; however, for DSS, a tendency close to the significance level was observed. The NRAS/BRAF status adversely affected survival: mutant NRAS was associated with a poorer PFS (*p* = 0.048) and OS (*p* = 0.037), while mutant BRAF was associated with a significantly more favorable OS (*p* = 0.045).

The results of the univariate model regarding SLN and mutational status were confirmed by Kaplan–Meier curves with log-rank tests. According to the SLN status, the comparison of PFS and DMFS revealed significant differences between SLN-negative and -positive cases (*p* = 0.001 and 0.004, respectively; Figure 1A,B). DSS showed nearly significant difference (*p* = 0.052): the 5-year disease-specific survival rate of SLN-negative cases was 92.7%, while that of SLN-positive cases was 77.5%. No significant difference of OS was found between the sentinel-positive and sentinel-negative groups. Evaluating the disease outcome of BRAF-, NRAS-mutant, and double-wild-type patients, NRAS-mutant cases showed less favorable outcomes in relation to almost every endpoints, while double-wild-type and BRAF-mutant cases showed a very similar survival pattern, with a more favorable prognosis (Figure 1C–F). Therefore, a comparison of merged BRAF-mutant/double-wild-type, and NRAS-mutant cases was performed, which showed statistically significant differences for PFS (*p* = 0.047) and OS (*p* = 0.035), while DMFS and DSS did not differ significantly. The 5-year PFS of BRAF-mutant/double-wild-type patients was 82.9%, whereas that of NRAS-positive cases was only 63.3%.

In multivariate analysis, except for the PFS, Breslow thickness still proved to be the strongest independent predictor of every endpoint. Compared to the univariate model, the predictive values of ulceration, mitosis, and invasion were weaker; mitosis was associated with DSS, and invasion with DMFS (Table 5). On the other hand, similarly to the univariate test, SLN positivity preserved the role of prediction on PFS (*p* = 0.005) and DMFS (*p* = 0.034). NRAS mutation proved to be a negative predictor of PFS (*p* = 0.047) and was nearly a significant predictor of DMFS (*p* = 0.06). In addition, patients’ age was an independent predictor of OS and DMFS, both in the univariate and in the multivariate model.

## 4. Discussion

In our retrospective study including 159 patients, the associations between routine clinico-pathological factors, mutational status of the primary tumors, and SLN status were assessed. According to numerous previously published results, the SLN status is considered one of the most important prognostic factors of melanoma [18,19,20], and our results also confirmed this theory. Progression and appearance of distant metastases showed an unfavorable outcome for SLN-positive patients. However, their effect on disease-specific and overall survival in our cohort might have been overwritten by numerous other factors, such as age pattern, comorbidities, and influence of innovative therapies. Generally, we consider that the evaluation of PFS and DMFS provides more informative results than that of DSS, since the influence of innovative therapies affects disease survival. In addition, disseminated disease was detected in the patients between 2012 and 2019, and in that period, the therapeutic modalities were very different according to the EMEA registration of medicines.

Nevertheless, the prognostic value of BRAF and NRAS mutation remains controversial. Some studies reported a shorter survival for BRAF-mutant stage IV melanoma patients [21,22], while other works did not find a significant difference for survival between patients with wild-type-BRAF and those with BRAF-mutant melanomas [23]. Among melanoma patients that received chemotherapy, circulating DNA of mutant BRAF was associated with a significantly worse overall survival compared to wild-type-BRAF patients, corresponding to 13 months and 30.6 months, respectively [24]. A meta-analysis of 59 heterogeneous studies (including 9243 patients) found a statistically significant association between reduced overall survival and the occurrence of BRAF mutation [25]. The prognostic value of NRAS mutation is also unclear [26,27]. A retrospective study of 217 patients confirmed that, compared to BRAF-mutant and -wild-type melanomas, NRAS-mutant tumors show a more aggressive biological behavior [14]. The latter corroborates our data, since the NRAS-mutant group showed a relatively unfavorable outcome compared to the BRAF-mutant and double-wild-type groups of patients. In addition, the BRAF-mutant and double-wild-type patients of our cohort showed a very similar survival pattern with respect to each applied endpoint.

Previously, only a few studies investigated the correlation between SLN status and oncogene mutation of the primary tumor. A case–control study of Manninen et al. enrolled 140 intermediate-thickness melanoma patients with or without SLN involvement, matched for age, gender, Breslow thickness, and location. They tested the common clinicopathological parameters (ulceration, mitotic count, and tumor regression), BRAF immunoreactivity, and cell motility involving actin-modulating proteins, in relation to SLN involvement and survival [28]. They found a significant correlation between SLN status and BRAF mutation; however, our study did not confirm their findings. A possible explanation of this contradiction is that, during the cited study, immunohistochemistry-based detection of V600E mutation was applied, while our group performed a more precise molecular analysis. It is of note that the indication of selective inhibitors is based on the latter method, and nowadays, immunohistochemical examination of BRAF is still an experimental rather than a reliable diagnostic option. Patient selection according to the experimental design showed another crucial difference, since they investigated exclusively patients with melanoma of intermediate thickness, while we did not include this prerequisite in our study. Nonetheless, we confirmed their other finding, that a higher mitotic activity of the primary tumor was related to SLN positivity. We also confirmed that SLN status was a strong independent predictive factor for progression-free survival and distant metastases-free survival. Another prospective cohort study suggested that melanoma patients with BRAF and NRAS mutation had an increased risk of tumor recurrence following a negative sentinel lymph node status. They concluded that, among melanoma patients with an earlier stage tumor, beside SLN negativity, intensive surveillance is required, taking into account BRAF and NRAS mutation status [29]. Without refusing their results, we found that the mutational status of NRAS has a higher impact on survival than that of BRAF. Furthermore, BRAF-mutant and double-wild-type patients showed a similar outcome. We found that beside Breslow thickness and sentinel lymph node status, NRAS mutation of the primary tumor is an independent predictor of PFS.

Moreover, our evaluations of BRAF-mutant, NRAS-mutant and double-wild-type tumors revealed that BRAF-positive patients were significantly younger, and NRAS-mutant primary tumors were thicker than BRAF-mutant and double-wild-type ones. Additionally, Breslow thickness, lymphovascular and/or perineural invasion, mitosis count, and ulceration of the primary tumor were associated with SLN status, while mutation of the primary tumor did not show any significant relationship with the SLN status in our study. Interestingly, in the SLN-positive cases, NRAS mutation of the primary tumor was less frequent than in SLN-negative cases (6.9% vs. 20.8%); however, patients with NRAS mutation showed a more unfavorable outcome. In a study investigating the prognostic and predictive values of oncogenic BRAF, NRAS, c-KIT, and MITF in melanoma, it was found that initial lymph node involvement was more frequent in patients with BRAF-mutated melanomas, than in cases presenting other mutations. This is partly similar to our results, as we found that SLN positivity was less frequent in NRAS-mutant, than in BRAF-mutant and wild-type cases [30].

Despite the low number of cases and the relatively heterogeneous population, our frequency analysis did not show statistically significant differences. Our results shares similarities with the study of Adler et al., where in comparison to wild-type primary tumors, BRAF and NRAS mutations in the primary tumor had a greater negative impact on the outcome of SLN-negative cases [29].

In conclusion, beside SLN positivity, age of the patients, Breslow thickness, lymphovascular invasion, and NRAS mutation of the primary tumor proved to be independent prognostic factors of melanoma progression. Though the age of the patients and Breslow thickness showed a significant relationship with the mutational status of the primary tumor by univariate analysis, age of the patients, NRAS mutation, and Breslow thickness remained independent prognostic factors. Therefore, despite the absence of positive SLN, the NRAS-positive patient subgroup requires closer monitoring to recognize disease progression. In contrast, the examination of our cohort did not confirm any significant association between BRAF mutation and SLN status and survival.

Our study has limitations. Melanomas are most frequently characterized by BRAF and NRAS mutation and, more rarely, by *c-KIT* oncogene mutations, but tumor suppressor genes such as *CDKN2A* and *PTEN* have an important role in the development and prognosis of melanoma [31]. In addition, TERT promoter mutation, detected in about half of melanoma cases, was independently associated with the prognosis of melanoma patients and of patients with other tumors [32].

The aim of our study was to evaluate the prognostic value of BRAF and NRAS mutation, evaluated in everyday clinical practice, in relation to the most important prognostic parameters, especially, the SLN status of melanoma patients.

## 5. Conclusions

In summary, independently of the SLN status, knowledge of the mutational status of primary melanoma lesions helps disease management and manifests additional benefits for the patients.

## Figures and Tables

**Figure 1 cancers-13-03302-f001:**
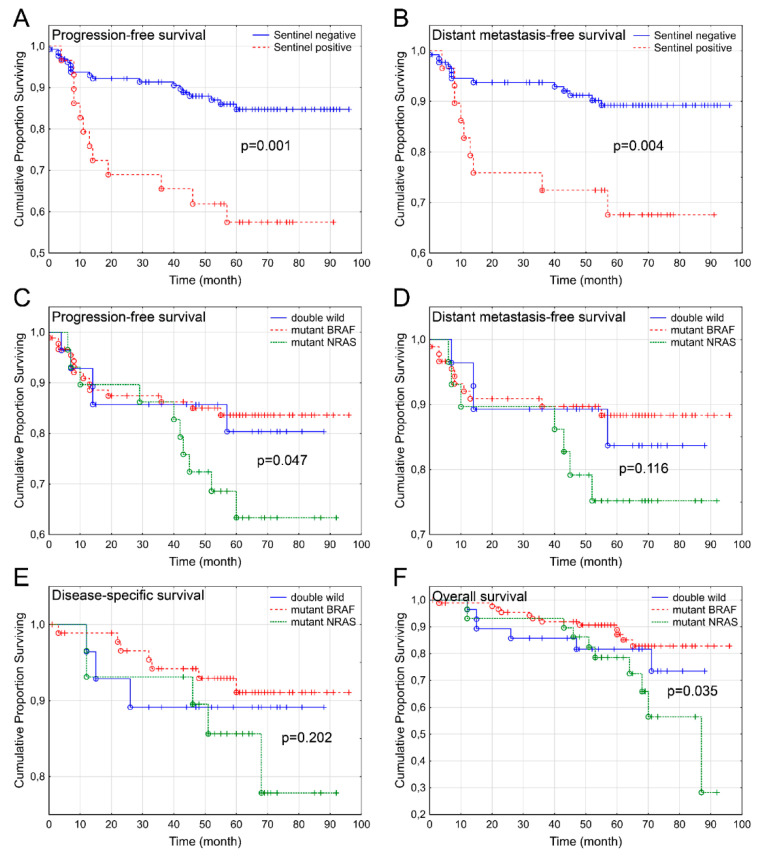
Kaplan–Meier curves of melanoma patients according to sentinel lymph node positivity and molecular subtypes. Sentinel lymph node positivity significantly impaired progression-free (**A**) and distant metastasis-free survival (**B**). Mutational status was analyzed in the primary tumor. Since the BRAF-mutant and double-wild-type subgroups showed a similar pattern, these categories were merged during the statistical analysis. Comparing the wild-type and NRAS-mutant groups, the latter showed a less favorable progression-free survival (**C**), while distant metastasis-free (**D**) and disease-specific (**E**) survival were not different. The overall survival (**F**) of NRAS-mutant patients showed a significantly worse outcome compared to that of the wild-type-NRAS subgroup. (o: Complete event; +: censored event).

**Table 1 cancers-13-03302-t001:** General characteristics of the studied patients with melanoma.

Characteristics	Values
All patients	159
Age (year) (median, min; max)	59 (18; 83)
Follow-up time (month) (median, min; max)	61 (1; 96)
Gender	
male	71 (44.7%)
female	88 (55.3)
Location	
upper extremity	35 (22%)
lower extremity	47 (29.6%)
trunk	77 (48.4%)
Breslow (mm) (median, min; max)	1.8 (0.51; 20)
Breslow categories	
I (≤1 mm)	21 (13.2%)
II (1.01–2.00 mm)	65 (40.9%)
III (2.01–4.00 mm)	62 (39%)
IV (>4 mm)	11 (6.9%)
Ulceration	
no	110 (69.2%)
yes	49 (30.8%)
Histological subtype	
SSM	124 (78%)
NM	31 (19.5%)
LMM	1 (0.6%)
ALM	1 (0.6%)
other	2 (1.3%)
Mitosis count (/mm^2^) (median, min; max)	4 (0; 31)
Invasion	
no	132 (83%)
vascular	11 (6.9%
lymphatic	11 (6.9%
vascular and lymphatic	2 (1.3%)
perineural	2 (1.3%)
NA	1 (0.6%)
TIL	
no	73 (45.9%)
yes	86 (54.1)
Regression	
no	79 (49.7%)
early	9 (5.7%)
late, <75%	50 (31.4%)
late, ≥75%	21 (13.2%)
Mutation categories	
double wild-type	28 (17.6%)
BRAF	90 (56.6%)
NRAS	29 (18.2%)
NA	12 (7.5%)
Sentinel status	
negative	130 (81.8%)
positive	29 (18.2)

SSM: superficial spreading melanoma; NM: nodular melanoma; LMM: lentigo malignant melanoma; ALM: acrolentiginous melanoma; TIL: tumor-infiltrating lymphocytes; NA: not available.

**Table 2 cancers-13-03302-t002:** Distributional differences of melanoma patients by sentinel lymph node status.

Characteristics	Sentinel Negative	Sentinel Positive	*p* (with Applied Statistics)
All patients	130	29	
Age (year) (median, min; max)	59.5 (18; 82)	58 (32; 83)	0.81 (Mann–Whitney)
Gender			0.208 (χ^2^)
male	55 (42.3%)	16 (55.2%)	
female	75 (57.7%)	13 (44.8%)	
Location			0.211 (χ^2^)
upper extremity	32 (24.6%)	3 (10.3%)	
lower extremity	36 (27.7%)	11 (37.9%)	
trunk	62 (47.7%)	15 (51.7%)	
Breslow (mm) (median, min; max)	1.64 (0.51; 20)	2.45 (0.79; 15)	**0.008 (Mann–Whitney)**
Ulceration			**0.007 (χ^2^)**
no	96 (73.8%)	14 (48.3%)	
yes	34 (26.2%)	15 (51.7%)	
Histological subtype			0.177 (χ^2^)
SSM	104 (80%)	20 (69%)	
NM	24 (18.5%)	7 (24.1%)	
other	2 (1.5%)	2 (6.9%)	
Mitosis count (/mm^2^) (median, min; max)	3 (0; 31)	5 (1; 30)	**0.009 (Mann–Whitney)**
Invasion (all type)			**0.004 (χ^2^)**
no	113 (86.7%)	19 (65.5%)	
yes	16 (12.3%)	10 (34.5%)	
NA	1 (0%)	0 (0%)	
TIL			0.34 (χ^2^)
no	62 (47.7%)	11 (37.9%)	
yes	68 (52.3%)	18 (62.1%)	
Regression			0.873 (χ^2^)
no	63 (48.5%)	16 (55.2%)	
early	7 (5.4%)	2 (6.9%)	
late, <75%	42 (32.3%)	8 (27.6%)	
late, ≥75%	18 (13.9%)	3 (10.3%)	
Mutation categories			0.218 (χ^2^)
double wild	23 (17.7%)	5 (17.2%)	
BRAF	71 (54.6%)	19 (65.5%)	
NRAS	27 (20.8%)	2 (6.9%)	
NA	9 (6.9%)	3 (10.3%)	

SSM: superficial spreading melanoma; NM: nodular melanoma; TIL: tumor-infiltrating lymphocytes; NA: not available. Bold means *p* < 0.05.

**Table 3 cancers-13-03302-t003:** Risk factors of sentinel node positivity by the binary logistic regression model.

Variable	OR (95% CI)	*p*
Age (≤59 vs. >59)	0.744 (0.268–2.065)	0.571
Gender (male vs. female)	0.563 (0.213–1.492)	0.248
Location (extremity vs. trunk)	1.142 (0.402–3.236)	0.804
Breslow (I–II vs. III–IV)	4.222 (1.201–14.837)	**0.025**
Ulceration (no vs. yes)	1.203 (0.383–3.788)	0.751
Mitosis (<4 vs. ≥4)	1.904 (0.617–5.874)	0.263
Invasion (no vs. yes)	1.759 (0.547–5.655)	0.343
TIL (no vs. yes)	1.185 (0.442–3.175)	0.736
BRAF (wild type vs. mutant)	1.287 (0.35–4.73)	0.704
NRAS (wild type vs. mutant)	0.314 (0.046–2.145)	0.237

OR: odds ratio; TIL: tumor-infiltrating lymphocytes. Bold means *p* < 0.05.

**Table 4 cancers-13-03302-t004:** Distributional differences of melanoma patients by molecular subtype (asterisk indicates a significant difference by the post hoc test).

Characteristics	Double Wild	BRAFmut	NRASmut	*p* (with Applied Statistics)
All patients	28	90	29	
Age (year) (median, min; max)	68 (19; 82)	53 (23; 83) *	66 (30; 81)	**0.001 (Kruskal–Wallis)**
Gender				0.555 (χ^2^)
male	12 (42.9%)	40 (44.4%)	16 (55.2%)	
female	16 (57.1%)	50 (55.6%)	13 (44.8%)	
Location				0.116 (χ^2^)
upper extremity	8 (28.6%)	16 (17.8%)	8 (27.6%)	
lower extremity	11 (39.3%)	28 (31.1%)	4 (13.8%)	
trunk	9 (31.1%)	46 (51.1%)	17 (58.6%)	
Breslow (mm) (median, min; max)	1.675 (0.73; 20)	1.64 (0.51; 15)	2.72 (0.84; 8) *	**0.018 (Kruskal–Wallis)**
Ulceration				0.534 (χ^2^)
no	21 (75%)	64 (71.1%)	18 (62.1%)	
yes	7 (25%)	26 (28.9%)	11 (37.9%)	
Histological subtype				0.244 (χ^2^)
SSM	23 (82.1%)	72 (80%)	20 (69%)	
NM	5 (17.9%)	15 (16.7%)	9 (24.1%)	
other	0 (0%)	3 (3.3%)	0 (0%)	
Mitosis count (/mm^2^) (median min; max)	3.5 (0; 26)	3 (0; 24)	4 (0; 31)	0.319 (Kruskal-Wallis)
Invasion (all type)				0.193 (χ^2^)
no	18 (64.3%)	78 (86.7%)	27 (93.1%)	
yes	10 (35.7%)	12 (13.3%)	1 (3.4%)	
NA	0 (0%)	0 (0%)	1 (3.4%)	
TIL				0.474 (χ^2^)
no	13 (46.4%)	38 (42.2%)	16 (55.2%)	
yes	15 (53.6%)	52 (67.8%)	13 (44.8%)	
Regression				0.873 (χ^2^)
no	15 (53.6%)	42 (46.7%)	16 (55.2%)	
early	2 (7.1%)	6 (6.7%)	0 (0%)	
late <75%	8 (28.6%)	30 (33.3%)	10 (34.5%)	
late ≥75%	3 (10.7%)	12 (13.3%)	3 (10.3%)	
Sentinel status				0.218 (χ^2^)
negative	23 (82.1%)	71 (78.9%)	27 (93.1%)	
positive	5 (17.9%)	19 (21.1%)	2 (6.9%)	

SSM: superficial spreading melanoma; NM: nodular melanoma; NA: not available; TIL: tumor-infiltrating lymphocytes. *: *p* < 0.05, compared to the other two groups; bold means *p* < 0.05.

**Table 5 cancers-13-03302-t005:** Cox univariate and multivariate analysis of various prognostic factors in melanoma patients, according to different survival endpoints (PFS: progression-free survival; DMFS: distant metastasis-free survival; DSS: disease-specific survival, OS: overall survival).

Parameters of Patients	Univariate Analysis		Multivariate Analysis	
	RR (95% CI)	*p*	RR (95% CI)	*p*
PFS				
Age	1.023 (0.996–1.051)	0.089	1.026 (0.995–1.058)	0.098
Gender (male vs. female)	1.084 (0.526–2.232)	0.827	1.259 (0.577–2.743)	0.563
Location (extremity vs. trunk)	0.786 (0.382–1.618)	0.513	1.117 (0.702–1.778)	0.641
Breslow thickness	1.206 (1.11–1.311)	**<10^−3^**	1.141 (1.018–1.28)	**0.024**
Ulceration (no vs. yes)	2.883 (1.406–5.912)	**0.004**	1.082 (0.398–2.937)	0.877
Mitosis number	1.081 (1.035–1.129)	**<10^−3^**	1.058 (0.997–1.123)	0.062
Invasion (no vs. yes)	2.534 (1.157–5.549)	**0.02**	1.8 (0.713–4.543)	0.111
TIL (no vs. yes)	1.018 (0.882–1.175)	0.812	1.07 (0.915–1.251)	0.213
NRAS (wild-type vs. mutant)	1.479 (1.003–2.18)	**0.048**	1.887 (1.008–3.53)	**0.047**
BRAF (wild-type vs. mutant)	0.577 (0.275–1.214)	0.148	0.719 (0.227–2.277)	0.575
Sentinel status (negative vs. positive)	3.315 (1.595–6.89)	**0.001**	3.301 (1.424–7.654)	**0.005**
DMFS				
Age	1.04 (1.005–1.075)	**0.023**	1.047 (1.007–1.087)	**0.019**
Gender (male vs. female)	1.197 (0.511–2.8)	0.679	1.39 (0.56–3.45)	0.477
Location (extremity vs. trunk)	0.873 (0.377–2.02)	0.75	0.979 (0.555–1.728)	0.942
Breslow thickness	1.223 (1.12–1.333)	**<10^−3^**	1.178 (1.039–1.336)	**0.011**
Ulceration (no vs. yes)	2.937 (1.268–6.802)	**0.012**	0.837 (0.236–2.964)	0.782
Mitosis number	1.089 (1.037–1.423)	**<10^−3^**	1.07 (0.999–1.146)	0.054
Invasion (no vs. yes)	4.143 (1.765–9.728)	**0.001**	3.255 (1.09–9.723)	**0.034**
TIL (no vs. yes)	0.966 (0.815–1.145)	0.687	1.046 (0.864–1.265)	0.647
NRAS (wild-type vs. mutant)	1.439 (0.911–2.274)	0.119	2.034 (0.97–4.265)	0.06
BRAF (wild-type vs. mutant)	0.569 (0.239–1.356)	0.203	0.643 (0.174–2.379)	0.508
Sentinel status (negative vs. positive)	3.288 (1.404–7.698)	**0.006**	2.929 (1.087–7.892)	**0.034**
DSS				
Age	1.041 (1–1.084)	0.051	1.047 (0.999–1.097)	0.056
Gender (male vs. female)	1.785 (0.62–5.142)	0.283	2.055 (0.629–6.713)	0.233
Location (extremity vs. trunk)	0.802 (0.299–2.154)	0.662	0.789 (0.396–1.571)	0.499
Breslow thickness	1.283 (1.161–1.418)	**<10^−3^**	1.294 (1.127–1.485)	**<10^−3^**
Ulceration (no vs. yes)	3.089 (1.5–8.3)	**0.025**	0.577 (0.113–2.948)	0.509
Mitosis number	1.108 (1.051–1.169)	**<10^−3^**	1.089 (1.006–1.177)	**0.034**
Invasion (no vs. yes)	4.346 (1.613–11.708)	**0.004**	3.485 (0.918–13.232)	0.067
TIL (no vs. yes)	1.032 (0.848–1.256)	0.751	1.086 (0.87–1.357)	0.466
NRAS (wild-type vs. mutant)	1.407 (0.821–2.41)	0.214	1.869 (0.757–4.615)	0.175
BRAF (wild-type vs. mutant)	0.544 (0.196–1.508)	0.242	0.771 (0.146–4.077)	0.759
Sentinel status (negative vs. positive)	2.593 (0.942–7.136)	0.065	1.774 (0.538–5.85)	0.346
OS				
Age	1.061 (1.026–1.097)	**<10^−3^**	1.068 (1.027–1.11)	**<10^−3^**
Gender (male vs. female)	0.741 (0.357–1.536)	0.42	1.17 (0.523–2.614)	0.703
Location (extremity vs. trunk)	0.961 (0.464–1.992)	0.915	0.743 (0.445–1.241)	0.256
Breslow thickness	1.296 (1.889–1.413)	**<10^−3^**	1.246 (1.112–1.397)	**<10^−3^**
Ulceration (no vs. yes)	3.95 (1.865–8.366)	**<10^−3^**	1.558 (0.541–4.482)	0.411
Mitosis number	1.098 (1.052–1.456)	**<10^−3^**	1.057 (0.995–1.123)	0.07
Invasion (no vs. yes)	2.165 (0.961–4.875)	0.062	1.399 (0.518–3.782)	0.508
TIL (no vs. yes)	1.012 (0.875–1.171)	0.87	1.06 (0.905–1.241)	0.472
NRAS (wild-type vs. mutant)	1.52 (1.026–2.253)	**0.037**	1.327 (0.704–2.5)	0.382
BRAF (wild-type vs. mutant)	0.456 (0.212–0.981)	**0.045**	1068 (0.336–3.398)	0.911
Sentinel status (negative vs. positive)	1.72 (0.761–3.89)	0.193	1.594 (0.641–3.962)	0.315

RR: relative risk; PFS: progression-free survival; DMFS: distant metastasis-free survival; DSS: disease-specific survival; OS: overall survival, TIL: tumor-infiltrating lymphocytes. Bold means *p* < 0.05

## Data Availability

Applied datasets contain personal data, therefore privacy issues do not allow sharing.

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
