# Peer review of "Predictive and Prognostic Value of BRAF and NRAS Mutation of 159 Sentinel Lymph Node Cases in Melanoma—A Retrospective Single-Institute Study"

_cancers, 2021, doi:10.3390/cancers13133302_

Round 1

Reviewer 1 Report

In the present study “Predictive value of BRAF and NRAS mutation of 159 sentinel lymph node cases in melanoma; a retrospective single institute study.”, the authors evaluate the correlation between known prognostic factors of melanoma, mutational occurrence of BRAF and NRAS in the primary tumor, sentinel lymph node status, and the association of these factors with disease outcome. In conclusion, beside the SLN positivity, Breslow thickness, lymphovascular invasion and NRAS mutation of the primary tumor proved to be independent prognostic factors of disease progression. Therefore, the study emphasizes the importance of identifying mutational status in the primary melanoma lesion for the better management of the patient.

In general, this manuscript is well-written, the methodology is elaborately described, and the results has been nicely discussed with appropriate references.

However, I have some suggestions which might improve the quality of presentation and its understanding.

Major Points:

  1. Please add limitations of the study in the discussion section. TERT promoter mutations are known to be associated with inferior outcome in melanoma. As such alterations were not included in the analysis, it can be added as limitation.
  2. I was wondering if microscopic satellitosis and surgical margin status was also included in the analysis.

Minor Points:

  1. If possible, please add in Figure 1, either as header or by the side of each curve/diagram, type of survival analysis/ different survival endpoints such as progression-free survival in 1(A) and distant metastasis-free survival in 1(B) and so on.
  2. Typo - (Pg 3, lane 95 - M&M section)—“ intermedier or high dose interferon” – Please correct intermediate.
  3. (Pg 4, lane 152 – Results section)—“ late regression in a lower rate than 75% occurred in 50 tumors” – Please rephrase this sentence. I think “late regression in less than 75% of tumor occurred in 50 tumors” sounds better.
  4. (Pg 4, lane 157 – Results section)—“ genotyping was not valid in 12 patients” – Please rephrase this sentence. I think “genotyping was not available in 12 patients” is more appropriate.
  5. Please consider elaborating discussion and references with some more recent similar studies such as “Prognostic and predictive values of oncogenic BRAF, NRAS, c-KIT and MITF in cutaneous and mucous melanoma”.

Author Response

Dear Reviewer 1.

Thank you for your time, work and suggestions for improving the quality of our manuscript.

Major points:

  1. Thank you for your very important notice, the TERT mutation was added as limitation of the study: See at the end of the manuscript. (Page, line, references)
  2. Microscopic satellitosis was not included in the analysis, because of the low number (less than 10) of cases. Primary melanomas were removed in two steps; 5 mm safety margin was completed to 1-2 cm according to the WHO guideline. (No insufficient margins were found in the population of patients.) See: (page 2, line 87)

Minor points:

  1. Corrected.
  2. Corrected.
  3. Corrected.
  4. Corrected.
  5. Discussion was completed with: In a study investigated the prognostic and predictive values of oncogenic BRAF, NRAS, c-KIT and MITF melanomas was found that initial lymph node involvement was more frequent in BRAF mutated melanomas, than in cases of other mutations. It was partly similar to our results, we found that Sentinel Lymph Node positivity was less frequent in NRAS, than in BRAF and wild type cases (Page: 11, line: 315)

Reviewer 2 Report

Liszkay et al investigates the correlation between SLN and known melanoma driver such as Braf and Nras using patients’ samples. They conclude that Nras mutant subgroup of patients requires close monitoring regardless SLN status. This is interesting observation that SLN cannot be a predictive marker for progression of Nras mutant melanoma. The manuscript is well documented with statistics. Two questions are raised before it’s published.

  1. Braf and Nras are major oncogenic drivers for melanoma initiation and development. However, only oncogenic activation is not sufficient to develop melanoma and second mutation in tumor suppressors is necessary such as Pten or CDKN2A. The authors did not describe or discuss this context. The association of Nras mutation with or without mutation of either Pten or CDKN2A with SLN should be assessed.

  1. Does SLN status correlate with progression of melanoma cells harboring only Braf mutation?

Author Response

Dear Reviewer 2.

Thank you for your time, work and suggestions for improving the quality of our manuscript.

Point 1.: Thank you for your recommendations.

Suggestion of the reviewer is very important, since despite mutations of BRAF or NRAS are the key mediators of melanoma, additional loss of tumor suppressors is necessary. According to previously published data, mutation of PTEN affects BRAF mutated or wild type BRAF/NRAS molecular subgroup in 40-60% (Shtivelman et al. Oncotarget . 2014 Apr 15;5(7):1701-52. doi: 10.18632/oncotarget.1892). Since these two groups dealt with the most favorable outcome, therefore we did not think that further categorization could add significant details to the big picture. Moreover, dividing into smaller subgroups may reduce general power of the statistical analysis, since our cohort contained only 29 cases with mutant NRAS. The same reason led us to miss the goal to analyze CDKN2A also.

Our aim was to compare sentinel lymph node status with the routinely applied parameters, and status of the mentioned tumor suppressors is not the part of that panel yet. Therefore, we thought that we had fulfilled the determined aim of the study.

Added to limitations: However, our study has limitations as well. Melanomas are characterized by the most frequently occurring BRAF and NRAS, more rarely c-KIT oncogene mutations, but tumor suppressor genes as CDKN2A and PTEN have an important role in the development and prognosis of melanoma. In addition TERT promoter mutation, detected in about half of melanoma cases, was independently associated with prognosis of melanoma and other tumors.

The aim of our study was to evaluate the prognostic value of the everyday clinical practice used BRAF and NRAS mutation in relationship with the most important prognostic parameters especially with the SLN status of melanoma. See: (Page 11, line: 334)

Point 2.: Thanks Reviewer 2 for the opportunity to clarify our findings. Further stratification of different molecular groups for SLN-positive and SLN-negative subset of patients revealed that among SLN-positive group progression-free survival did not differ between BRAF, NRAS and double wild type group. In addition, according to progression among SLN-negative group, NRAS-mutant subgroup showed the most unfavorable outcome significantly, which corroborates our major results of the manuscript.

Reviewer 3 Report

The manuscript "Predictive value of BRAF and NRAS mutation of 159 sentinel lymph node cases in melanoma; a retrospective single institute study" presented by Liszkay et al. described a retrospective study on 159 melanoma patients who were tested for positive or negative SLN. The major findings of this works confirm previous reports in melanoma that the Breslow thickness is the single most predictive marker in melanoma. However, the uniqueness of the cohort add value to the study.

Reviewer's comments

1- The figure 1 will benefit from adding to each panel what is calculated: overall survival (OS), disease-specific survival (DSS), progression-free survival (PFS) or distant metastasis-free survival (DMFS), in addition to the figure legend.

2- In the discussion section the authors don't discuss the age of the patient and the relation with braf mutation and the different end points survival. The manuscript will benefit from adding some discussion.

3- The authors should discuss and present survival data relative to the treatment the patients received. This was only mention in the method part and the authors do not provide any data or discussion regarding the influence of the treatment on the survival.

Overall the manuscript was well written and the statistical methods applied were adequate. The title does not have a good match with the results and the main conclusions.

Author Response

Dear Reviewer 3.

Thank you for your time, work and suggestions for improving the quality of our manuscript.

  1. Corrected.
  2. “In conclusion beside the importance of SLN positivity, age of patients…” was added to the discussion. (Thank you for your notice.) Furthermore it was added to the conclusion. “Though the age of patients and Breslow-thickness showed significant relationship with the mutational status of primary tumor by univariate analysis, however, age of patients, NRAS mutation and Breslow-thickness remained independent prognostic factors.” (See: page 11, line 325)
  3. In addition disseminated disease of the patients occurred between 2012 and 2019 and in that period the therapeutic modalities were very different according to the EMEA registration of medicines (until 2014 no combinated targeted therapy, until 2015 no anti-PD1 therapy was registrated). We added to the discussion part paragraph one.

The title was changed “Predictive and prognostic value of BRAF and NRAS mutation of 159 sentinel lymph node cases in melanoma; retrospective single institute study.”

Round 2

Reviewer 2 Report

The authors addressed the reviewer's questions.